# Stormwater Treatment Using Natural and Engineered Options in an Urban Growth Area: A Case Study in the West of Melbourne

Peter Sanciolo [1,*], Ashok K. Sharma [1,2,*] , Dimuth Navaratna [1,2] and Shobha Muthukumaran [1,2,*]

1   Institute for Sustainable Industries & Liveable Cities, Victoria University, Melbourne, VIC 3011, Australia; dimuth.navaratna@vu.edu.au
2   College of Sport, Health & Engineering, Victoria University, Melbourne, VIC 3011, Australia
*   Correspondence: peter.sanciolo@vu.edu.au (P.S.); ashok.sharma@vu.edu.au (A.K.S.); shobha.muthukumaran@vu.edu.au (S.M.)

**Abstract:** The expected increase in urbanization and population in coming years is going to increase the impervious land area, leading to substantial increases in stormwater runoff and hydrological challenges, and presents significant challenges for urban potable water supply. These are worldwide challenges that can potentially be ameliorated by harvesting stormwater for potable use or for other uses that can reduce the pressure on potable water supply. This study sought to assist the local water authority in planning for future potable water supply through a review of the scientific literature to determine the likely chemical and microbial characteristics of stormwater, the treatment train (TT) requirements, and the likely costs of treatment to achieve potable standards for the high-growth metropolitan region of Melbourne, Australia. Literature stormwater quality statistical data and treatment process performance data were used to model the expected product water microbial and chemical quality after treatment using a number of advanced TT options. The results of the modelling were compared with literature microbial log reduction targets (LRTs) for the potable use of stormwater and with the Australian Drinking Water Guidelines (ADWG). It was found that a reverse osmosis (RO)-based TT with microfiltration pre-treatment and post-RO advanced oxidation and chlorination in storage reservoirs is a conservative stormwater potable use treatment option. A less conservative and less expensive ozone-and-biologically active filtration ($O_3$/BAF)-based TT option is also proposed if RO concentrate disposal is deemed to be too challenging. These results could be useful in climate change adaptation involving the evaluation of options for the mitigation of future population-growth- and climate-change-driven water supply challenges, as well as urbanization-driven stormwater hydrology and receiving water pollution challenges.

**Keywords:** stormwater harvesting; potable water supply; potable reuse; climate change; treatment trains; stormwater quality





## 1. Introduction

Many modern cities are undergoing rapid urbanization. The expected growth in population coupled with a changing climate in the west of Melbourne, Australia, for example, is projected to present significant challenges related to stormwater runoff generated from impervious surfaces and increased demand for potable water supply. The metropolitan region of Greater Melbourne has five major waterway catchments. The major catchment in the west of Melbourne, the Werribee catchment, is undergoing rapid population growth. It had an estimated population of 614,000 people in 2016 and is predicted to grow to 1.41 million by 2050. Several municipalities in the region have transitioned from primarily rural to primarily urban areas [1]. The increased stormwater runoff from increased land coverage of impervious surfaces stands to degrade downstream waterways due to a combination of high volumes and poor water quality. While stormwater runoff is expected to be partially

mitigated by anticipated drier conditions due to climate change, water supply challenges are likely to be magnified by drier climatic conditions. This is based on the inference of the DELWP guidelines [2] on the impact of current rainfall and PET changes under low, medium, and high climate changes. However, Wiwoho et al. [3] have highlighted an increase in rainfall amount, duration, and extreme rain occurrences, and a decrease in light rain occurrences in some parts of the world. Detailed stormwater runoff quality and quantity modelling for this case study area are reported in the literature by Sharma et al. [4].

This research investigated the feasibility of harvesting and treating stormwater from the expected development in the west of Melbourne, Australia, to address the expected potable water supply challenges associated with a drier climate and a growing population. There is wide variability in stormwater quality in urban environments due to catchment characteristics (e.g., rural, urban, residential, agricultural, industrial), the activities within catchments, and the conditions of sewerage infrastructure [5]. Stormwater quality also depends on when water quality is analysed and reported. The levels of lead in Australian stormwater, for example, are expected to be lower now than 20 years ago due to the phasing out of leaded petrol in 2002. The level of pathogens and chemical pollutants also depends on build-up during dry periods and wash-off during and after rainfall events.

The safety of ingestion of recycled water involves the performance of a quantitative microbial risk assessment (QMRA). This involves detailed knowledge of the microbial quality of water, allowing the estimation of the annual probability of infection and the disability-adjusted life years (DALYs) to be calculated for assumed ingestion of recycled water. In Australia, the health-based drinking water benchmarks for infection and disability-adjusted life years (DALYs) are $1 \times 10^{-4}$ and $1 \times 10^{-6}$, respectively. The QMRA can then be used to determine the log reduction target (LRT) for the treatment of stormwater. "Australian Guidelines for Water Recycling (AGWR)—Phase 2—Augmentation of Potable Water supplies" [6] states that default LRT values are not available for stormwater due to the variability in stormwater quality and the influence of catchment characteristics. It only gives default LRTs for the treatment of municipal wastewater for potable use.

Due to the lack of guideline default stormwater treatment LRTs for potable use and the high cost of extensive, site-specific, and temporal analysis of the microbial and chemical quality of stormwater, harvested stormwater in Australia has primarily been constrained to non-potable applications, such as irrigation of sports fields, golf courses, parks, and gardens. Consequently, Australian case studies on the potable use of stormwater are very few. One case study on indirect stormwater potable use is that of Orange Council, NSW, [7], which treated harvested stormwater for use to top up water supply storages. One notable case study on direct potable use of stormwater is the Yarra Valley Water Kalkallo Project in the north of Melbourne [8]. However, this is currently a proof-of-concept project, with the most recent literature [8] on the project stating that a treatment plant to treat stormwater has been built but is yet to be operated because of delays in the wider development that have meant that adequate stormwater runoff quantities are not yet available.

There are, however, many international examples of wastewater potable reuse schemes. Two typical treatment trains (TTs) that are commonly considered for potable reuse of municipal wastewater are the following [9]:

1. A reverse osmosis (RO)-based treatment process primarily consisting of microfiltration (MF), RO, advanced oxidation (UV/H$_2$O$_2$), and chlorination in a storage reservoir (Cl$_2$).
2. An ozone-and-biologically active filtration (O$_3$/BAF)-based process primarily consisting of O$_3$/BAF, ultrafiltration (UF), advanced oxidation (UV/H$_2$O$_2$), and chlorination in a storage reservoir.

Regardless of whether a direct potable reuse scheme (DPR) uses stormwater or wastewater, the rationale for TT design is to aim for a TT that meets the microbial potable use LRTs and guideline chemical drinking water standards. In the preliminary design stage, each stage in the TT is assigned an accepted default microbial log reduction value (LRV) such

that the sum LRV of all stages in the TT exceeds the potable reuse LRTs for each pathogen of concern in feedwater. Additionally, the TT must also be able to reduce the chemical constituents to the safe levels dictated by drinking water regulations.

To support the development of microbial LRTs for the potable use of stormwater, Schoen et al. [10] performed a QMRA for simulated stormwater that had been contaminated to different degrees by raw municipal wastewater (secondary effluent). They conducted a probabilistic QMRA to derive the pathogen LRTs that corresponded to a benchmark infection risk of $10^{-4}$ per person per year (ppy). This infection risk roughly corresponds to the WHO tolerable burden of disease of $10^{-6}$ disability-adjusted life years (DALYs) ppy for drinking treated stormwater. They compared their LRTs for two simulated stormwater samples consisting of different dilutions of municipal wastewater with stormwater (1:10, 1:1000) and concluded that the LRTs for the 1:10-diluted simulated wastewater could be considered conservative LRTs for stormwater.

In the absence of site-specific stormwater quality data and LRTs for potable use for the west of Melbourne, the current research aims to assist in the planning for future stormwater potable use by comparing the expected performance of two typically considered wastewater potable reuse TTs to assess the relative strengths and weaknesses of the two TT types for stormwater treatment. It compares the expected LRVs for an RO-based TT and an ozone–biologically active filtration ($O_3$/BAF)-based TT with the indicative LRT estimates by Schoen et al. [10] for the potable use of stormwater. It also models the expected product water chemical quality using literature-sourced chemical stormwater quality and efficacy estimates for the individual process stages of the two TTs and compares the treated-water quality to the ADWG values. Indicative cost estimates are also made based on literature-reported costings for similar TTs. It is hoped that the findings of this study can be used to facilitate the high-level comparison of stormwater treatment options and other potential options, such as minimisation of leaks in the reticulation system or the pumping of water from the local seawater desalination plant, to overcome the expected potable water supply challenges associated with a drier climate and a growing population in the west of Melbourne.

## 2. Methods

The overarching rationale for this desktop study is to test the expected performance of TTs similar to those commonly adopted in wastewater DPR and IPR schemes in the United States and internationally against recent literature LRTs for the potable reuse of stormwater [10] and the ADWG using stormwater quality data from the AGWR [5] to determine if these TTs would be appropriate for stormwater treatment in the west of Melbourne.

### 2.1. Treatment Trains

The conceptual design of TTs for the potable use/reuse of stormwater was adapted from existing typical examples of TTs for the potable reuse of wastewater. Wetlands were used as the first stage of the TTs, as these are convenient storage reservoirs for stormwater prior to treatment. They also offer removal of organic and inorganic contaminants through a combination of microbial degradation, phytodegradation, adsorption, and sedimentation [11]. The selection of subsequent stages in the TTs was based on two of the three examples of typical wastewater potable reuse TTs outlined in the Water Reuse Foundation Framework for DPR [9]. They were selected due to their frequent use in potable reuse schemes and for the availability of an evaluation of the microbial risk associated with similar treatment trains in the scientific literature [12]. Similar TTs are also featured and discussed in the WHO Potable Reuse Guidance for Producing Safe Drinking Water [13]. The four chosen conceptual designs for the TTs are shown in Figure 1.

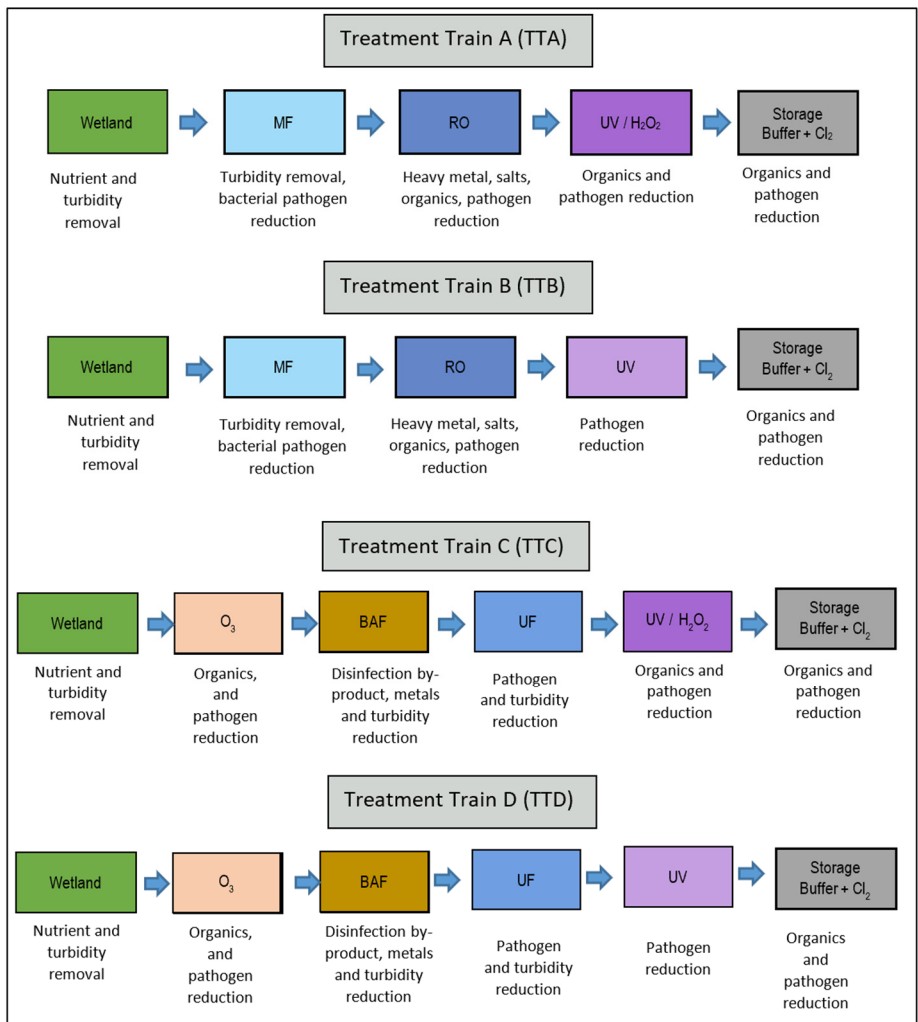

**Figure 1.** Treatment trains.

TTA is a modified version of a TT investigated by Soller et al. (TT1) [12]. This is also similar to the TT used in the full-scale California facility [14]. In the current project modelling, wetland treatment was added at the start to assist in purification and in stormwater management. The rationale for selecting this TTA was that if this TT allows WWTP wastewater to achieve microbial drinking water standards, it should be more than enough treatment for less polluted stormwater. This TT uses UV and $H_2O_2$ at very high energy (800 mJ/cm$^2$), which is much more than what is required for disinfection. The most common purpose of this treatment is the destruction of trace organic toxins, such as NDMA and 1,4-dioxane [12].

TTB is the same as TTA except that conventional UV disinfection is used instead of the advanced oxidation stage (UV/$H_2O_2$). This process train is a modified version of the TT1b by Soller et al. [12]. The rationale for selecting this TT was that low-dose-UV treatment without hydrogen peroxide addition may be sufficient, as stormwater is less contaminated than WWTP wastewater.

TTC is a modified version of the TT3a proposed by Soller et al. [12]. This TT has been proposed as an alternative TT for inland areas where the challenge of safe disposal of RO concentrate is a major impediment to the adoption of RO technology [9]. The rationale for selecting this TT was that if this TT allows WWTP wastewater to achieve microbial drinking water standards, it should be more than enough treatment for less polluted stormwater.

TTD is the same as TTC except that conventional UV disinfection is used instead of the advanced oxidation stage (UV/$H_2O_2$). This process train is a modified version of the TT3b

by Soller et al. [12]. As with TTB, the rationale for selecting this TT was that low-dose-UV treatment without hydrogen peroxide addition may be sufficient, as stormwater is less contaminated than WWTP wastewater.

## 2.2. Estimation of Key Microbial Parameter Removal

The microbial removal performance of the four TTs was estimated by adding the default removal credits for the individual process stages from the literature (see Table 1).

**Table 1.** Literature LRV assignments used to calculate the total treatment train LRVs.

| Process Stage | Bacteria (*Campylobacter*) | Viruses (Adenovirus) | *Cryptosporidium* | Reference |
|---|---|---|---|---|
| Wetland | 0.2 | 0 | 0 | [15] |
| MF | 3 | 1 | 3 | [16] |
| RO | 3 | 3 | 3 | [13] [a] |
| UV (12 mJ/cm$^2$) | 4 | 1 | 2 | [12] |
| UV/H$_2$O$_2$ (800 mJ/cm$^2$) | 4 | 4 | 4 | [9,12] [b] |
| O$_3$ + BAF | 4 | 4 | 1 | [12] |
| UF | 3 | 2.5 | 3 | [16] |
| Cl$_2$ (Ct > 15 mg/L·min) | 4 | 4 | 0 | [16] |

Note: [a] Values used for Case Study 5: Perth, Australia, groundwater replenishment ([13] Table CS5.1). [b] Reference state 6-log reduction for bacteria, viruses, and *Cryptosporidium*. Credits reduced here to a maximum of 4 as per principles of good practice design stated in [16].

It is important to note that the RO stage in TTA and TTB can achieve greater than 6-log reductions in microbial pathogens, but operational monitoring lacks sensitivity and reduces the log credits that can be claimed to 2–4 logs [13]. Credits are commonly reduced to a maximum of 4 as per principles of good practice design as stated by the WSAA manual [16]. A median value of 3 was used in the modelling.

The total LRVs of the TTs were compared to Schoen et al.'s [10] stormwater LRTs for potable use and the AGWR's [6] default LRTs for the potable reuse of wastewater.

## 2.3. Estimation of Key Chemical Parameter Removal

The estimation of microbial parameter removal was undertaken through the consecutive application of the percentage removal in each stage of the TTs to the initial stormwater composition with regards to key parameters. Due to the lack of available data pertaining stormwater in the case study area, the Werribee catchment in the west of Melbourne, the initial stormwater composition was assumed to be that given in the collation of literature data in the Stormwater Harvesting Guidelines [5]. The 95th percentile statistical summary chemical water quality data from these Stormwater Harvesting Guidelines was compared to the ADWG values [17], and the concentration of the parameters that exceeded or were close to the ADWG values were estimated: polyaromatic hydrocarbons (PAH), As, Cd, and Pb. Biological oxygen demand (BOD) removal and chemical oxygen demand (COD) removal were also estimated as general indicators of water quality.

Literature advanced treatment process stage removal dealing with wastewater was used, as very few literature studies dealing with stormwater treatment were found. The literature studies used to estimate removal in each TT process stage for each selected chemical are summarized in Table 2. Details of the raw water type and the treatment conditions for the literature references are shown in Table 3.

**Table 2.** Percentage chemical parameter removal and literature references used to estimate TT performance.

| Process Stage | Parameter | | | | | |
|---|---|---|---|---|---|---|
| | PAHs (Benzo(a)pyrene) | As | Cd | Pb | BOD | COD |
| Wetlands | 68 [A] | 30 [B] | 30 [B] | 30 [B] | 89 [C] | 72 [C] |
| MF | 0 | 0 | 0 | 0 | 0 | 0 |
| RO | 80 [D] | 85 [E] | 98 [E] | 95 [F] | 99 [G] | 99 [G] |
| UV | 0 [H] | 0 | 0 | 0 | 0 [I] | 0 [I] |
| UV/$H_2O_2$ | 99 [J] | 0 | 0 | 0 | 0 [I] | 0 [I] |
| $O_3$ | * | 0 | 0 | 0 | 67[11] [K] | 64[11] [K] |
| BAF | 97 [L] | 99 [M] | 86 [N] | 95 [N] | - | - |
| UF | 67 [O] | 0 [P] | 0 [P] | 0 [P] | 97 [G] | 97 [G] |
| Chlorination in engineered storage buffer | 80 [D] | 0 | 0 | 0 | 30 [Q] | 10 [R] |

Note: (A): Fountoulakis 2009 [18]; (B): Haarstad 2012 [19]; (C): Varma 2021 [11]; (D): EPHC, NHMRC, NR-MMC 2008 [6]; (E): USEPA 1988 [20]; (F): Ozbey-Unal 2020 [21], (G): Jadhao 2012 [22]; (H): Sanches 2011 [23]; (I): Muhammad 2008 [24]; (J): Rubio Clemente 2018 [25]; (K): He 2013 [26]; (L): Augulyte et al., 2009 [27]; (M): Pokhrel et al., 2009 [28]; (N): Dong et al., 2018 [29]; (O): Smol et al., 2012 [30]; (P): Kumar et al., 2022 [31]; (Q): Ishihara et al., 2009 [32]; (R): Ustun et al., 2011 [33], * the combined removal for O3 and BAF of reference (L) was used rather the individual $O_3$ removal of PAH.

## 2.3.1. PAH Removal

The vast majority of PAH content in stormwater is removed in the wetland and RO stages (TTA and TTB) or in the wetland and $O_3$/BAF stages (TTC and TTD). The advanced oxidation (TTA and TTC), UF (TTC and TTD), and chlorination (all TTs) stages act as polishing stages.

Wetland treatment: The modelling in the current study used a PAH removal rate of 68% for the free-water-surface constructed wetland [18]. The PAH removal attributed to constructed wetlands in the current modelling was based on a study conducted in China, Greece (35.3° north of the equator) [18]. This study was selected due to the similarity in distance from the equator to Melbourne (37.5° south of the equator). The efficiency of the removal of polycyclic aromatic compounds (PAHs) was evaluated in a pilot-scale constructed wetland system combining a free-water-surface wetland, a subsurface wetland, and a gravel filter in parallel. The average PAH removal rates were 79.2% for the subsurface-flow constructed wetland, 68.2% for the free-water-surface constructed wetland, and 73.3% for the gravel filter, respectively.

RO treatment: The modelling used a conservative value of 80% for RO removal of PAH (benzo(a)pyrene) [6]. Reverse osmosis has been described as one of the, if not the, most effective single-unit process for the removal of chemicals of concern in water treatment. It typically removes >90% and often >99% of wastewater organics, depending on the compound. The indicative RO removal of benz(a)pyrene (3,4-benzopyrene), a polyaromatic hydrocarbon (PAH), is given, in the AGWR—Augmentation of Potable Water Supplies [6]—as >80%. The RO rejection of smaller polyaromatic hydrocarbons, naphthalene, anthracene, and phenathrene, was found to be 98–99% (USEPA 1987) [34]. Similarly, the rates of RO removal of other polyaromatic hydrocarbons, such as acenanthrene, fluoranthene, naphthalene, and phenthrene, of 99%, 86%, 99%, and 99%, respectively, were reported [35].

**Table 3.** Literature treatment conditions.

| Treatment Train Stage | Chemical | Water Type | Key Conditions | Comment and Removal | Reference and Reference Type |
|---|---|---|---|---|---|
| Wetland | PAH | Stormwater | Feed concentration of $786 \pm 514$ ng/L; 42 m$^2$ wetland; southern Greece (35°19__N and 25°10__E); mixed cultures of two species of reed, Phragmites australis and Arudo donax; mean water temperature: 12.1 and 34.1 °C | Free-water-surface constructed wetland; 68% removal | [18], experimental |
| Wetland | Heavy metals | Stormwater | Various constructions and conditions | Typical removal: 30–60% removal, 30% used in modelling | [19], literature review |
| Wetland | BOD, COD | Stormwater | No media, with floating, submerged, and emergent plants; continuous water supply | Free-water-surface constructed wetland; around 89% BOD removal and 72% COD removal | [11], literature review |
| RO | PAH | Wastewater | Validation of specific application and operational conditions required | Indicative, >80% removal; intended to be informative and not to be used as the design basis for schemes | [6] (Table 4.10 of water recycling guidelines) |
| RO | As(III) | Synthetic brackish water | Feed concentration between 0.36 and 1.2 mg/L As(III), spiked drinking water, Dow 5K membranes, tests at manufacturer operating specifications | 85% removal | [20] |
| RO | Cd | Drinking water | Feed concentrations between 0.47 mg/L and 1.9 mg/L, spiked drinking water, Dow 5K membranes, tests at manufacturer operating specifications | 98% removal | [20] |
| RO | Pb | Municipal and industrial wastewater | Laboratory and on-site pilot-scale tests, feed concentration of 1.5 mg/L, various conditions | 89 to 100% removal, 95% used in modelling | [21], experimental |
| RO | COD, BOD | Hospital wastewater | Feed concentrations: 200–235 mg/L COD, 95–115 mg/L BOD; feed rate: 10–14 L/h; variable pressure to 13.6 Bar maximum; specific flux: 90–190 L/m$^2$/h/bar | More than 99% removal for both COD and BOD | [22], experimental |
| UV | PAH | Natural water | Feed concentration: 3.9 to 5.6 µg/L; 3 different PAHs; 3 different water matrices; fluence between 40 and 1500 mJ/cm$^2$ | Negligible PAH removal at 40 mJ/cm$^2$ | [23], experimental |
| UV | COD, BOD | Raw and biotreated textile dye bath effluent | Feed concentrations: 760 mg/L COD, 261 mg/L BOD; UV dose: 5 mW/cm$^2$; exposure time between 5 and 25 min | Negligible COD and BOD removal at 12 mJ/cm$^2$, 35% BOD and 25% COD removal at 7500 mJ/cm$^2$ | [24], experimental |

**Table 3.** *Cont.*

| Treatment Train Stage | Chemical | Water Type | Key Conditions | Comment and Removal | Reference and Reference Type |
|---|---|---|---|---|---|
| $UV/H_2O_2$ | PAH | Natural water | Feed concentration: 3 µg/L; 30 min contact time; UV radiation: 170 µW/cm$^2$; 10 mg/L $H_2O_2$ | 99% removal at 306 mJ/cm$^2$ | [25], experimental |
| $UV/H_2O_2$ | COD, BOD | Raw and biotreated textile dye bath effluent | Feed concentrations: 760 mg/L COD, 261 mg/L BOD; UV intensity: 5 mW/cm$^2$ and 254 nm; 150–200 mg/L $H_2O_2$ for raw wastewater, 100–150 mg/L for biotreated wastewater | Negligible BOD and COD removal at 800 mJ/cm$^2$ Raw wastewater: 35% COD removal, 44% BOD removal at 7500 mJ/cm$^2$ Biotreated: ~85% COD removal and ~90% BOD removal at 7500 mJ/cm$^2$ | [24], experimental |
| $Cl_2$ | PAH | Wastewater | Various conditions | Indicative, >80% removal | [6] |
| $Cl_2$ | COD | Industrial wastewater | Feed concentration: 39 mg/L; chlorination after coagulation and flocculation; 1.2 mg/L free chlorine; 30 min contact time | 10% removal | [33], experimental |
| $Cl_2$ | BOD | Secondary-effluent wastewater | Various feed concentrations: 12–30 mg/L, 5 mg/L residual $Cl_2$; 15 min contact time | 67% to 20% depending on starting concentration; conservative setting of 30% removal chosen for modelling | [32], experimental |
| $O_3$ + biological treatment | PAH | Contaminated water | Feed PAH concentration: ~5000 µg/L; 0.5 mg/L ozone; 30 min ozone treatment; 24 h biological treatment in flask | 91% PAH removal overall | [36], experimental |
| BAC | PAH | Diesel and petrol Synthetic wastewater | Petroleum content: 5 mg/L; ~1100 µg/L PAH; 8 L BAC to 300 L contaminated water; aerobic conditions; 12–24 h contact time | 97% PAH removal | [27], experimental |
| $O_3$ + BAC | As | Groundwater | Feed As concentration: 14–27 µg/L; 43 min contact time | 99% removal | [28], experimental |
| BAC | Pb, Cd | Wastewater | Feed Pb and Cd concentrations: ~ 200 µg/L; 50–150 mg/L activated carbon; 2 h contact time | 99% Pb and 86% Cd removal | [29], experimental |
| $O_3$ | COD, BOD | Secondary-effluent wastewater | Full scale, variable feed concentrations (~10–80 mgO$_2$/L COD, ~2–10 mgO$_2$/L BOD), 11–13 mg/L ozone | 8%–88% COD removal with most results in 10–20% removal range, ~0% BOD removal | [37], experimental |
| $O_3$ + BAF | COD | Surface water | Biological sand filter, full-scale plant, 30–60 min contact time, 17 mg/L $O_3$ concentration | Two different plants: one achieved ~50% COD removal, the other, ~20% COD removal | [38], experimental |
| $O_3$ + BAF | BOD, COD | Textile effluent | Biological aerated filtration; feed COD ≤ 110 mg/L, BOD ≤ 30 mg/L; 20–25 mg/L ozone dose; 3.3 h hydraulic retention time; 6 air-to-water flow ratio | Approximately 64% COD removal, 67% BOD removal | [26], experimental |

Table 3. *Cont.*

| Treatment Train Stage | Chemical | Water Type | Key Conditions | Comment and Removal | Reference and Reference Type |
|---|---|---|---|---|---|
| UF | PAH | Biologically treated wastewater | Feed PAH concentration: 22–38 μg/L; 0.04 μm pore size | 67% removal | [30], experimental |
| UF | COD, BOD | Hospital wastewater | Feed concentrations: 200–235 mg/L COD, 95–115 mg/L BOD; 0.01 μm pore size (1 kDa molecular-weight cutoff) | 97% removal for both COD and BOD | [22], experimental |
| | COD, BOD | Stormwater | Feed concentrations: 11–32 mg/L BOD, 28–60 mg/L COD; 50 kDa molecular-weight cutoff UF | | [39], experimental |

RO, however, cannot be relied upon as the only process stage, as it is inefficient in the removal of low-molecular-weight organics, such as formic acid, methanol, formaldehyde, and urea. It also performs poorly in the rejection of boron [40]. Reverse osmosis rejection of neutral organic pollutants has been found to increase with the increase in compound length and width and to decrease with the increase in compound hydrophobicity [41]. Some non-polar, low-molecular-weight organics, such as N-Nitroso-dimethylamine (NDMA) and 1,4-dioxane, can pass through RO membranes [42]. Another disadvantage of RO treatment is the challenge of the disposal of the concentrate. For potentially very harmful organics such as PAHs, ocean disposal would require further treatment, and surface-water disposal in inland regions is likely to be prohibited. $O_3$/BAF TTs may be more suited to potable reuse in inland regions.

UV treatment: The modelling used a PAH (benzo(a)pyrene) removal rate of 0% for UV alone. The UV dose used in the modelling of the effect of UV treatment was 12 mJ/cm$^2$, as this is a dose that is consistent with traditional wastewater disinfection and has been used in the literature to estimate the risk of potable reuse [12]. Little or no PAH degradation is expected at this low fluence. Sanches et al. [23] investigated the effect of UV dose on the degradation of three different PAHs (anthracene, fluoranthene, and benz(a)pyrene) in three different matrices (laboratory groundwater, groundwater, and surface water) and found very low removal at these low UV doses. PAH removal was found to require high fluence and was found to depend on the water matrix. Approximately 10% benzo(a)pyrene reduction in the groundwater types and approximately 0% reduction in surface water were reported at the fluence of 40 mJ/cm$^2$. Less degradation was reported for the other PAHs. Using much higher UV fluence, 1500 mJ/cm$^2$, anthracene and benzo(a)pyrene were efficiently degraded, with much higher percent removal being obtained when present in groundwater (83–93%) compared with surface water (36–48%). The removal percentages obtained for fluoranthene were lower and ranged from 13 to 54% in the different water matrices tested.

$UV/H_2O_2$ treatment: The modelling at the fluence of 800 mJ/cm$^2$ used a PAH (benzo(a)pyrene) removal rate of 99% for $UV/H_2O_2$. A study on the $UV/H_2O_2$ treatment of natural water by Rubio Clemente [25] found that anthracene and benzo(a)pyrene removal rates of approximately 88% and 78%, respectively, were achieved using $H_2O_2$ at 10 mg/L and irradiance of 0.17 mW/cm$^2$ for 1 min, equating to the fluence of 10.2 mJ/cm$^2$. Increasing the exposure time to 30 min, equal to the fluence of 306 mJ/cm$^2$, increased the removal of the two PAHs to approximately 99%.

$O_3$/BAF treatment: The modelling in the current study used a 97% PAH removal rate for the $O_3$/BAF combination [27]. Ozonation followed by biological treatment has been found to be very effective for PAH removal. An overall PAH (benzo(a)pyrene) removal rate of 91% after 30 min ozonation at 0.5 mg/L $O_3$ and 24 h biotreatment in stirred flasks was achieved [36]. Degradation using ozone alone under the same dose/time regime

only achieved 63% degradation. The use of BAF rather than stirred flasks is expected to yield timelier PAH removal. Overall PAH removal efficiency rates of 96.9% to 99.7% were achieved within 24 h. The major contributor to removal was sorption rather than biodegradation [27].

UF treatment: The modelling in the current study used 67% PAH removal for UF treatment [30]. $O_3$/BAF-based TTs (TTC and TTD) heavily rely on ultrafiltration (UF) for PAH removal. Smol and Wlodarczk-Mekula [30] investigated the use of ultrafiltration to remove PAHs from highly polluted water from a coke process. The total concentration of 16 PAHs in the process of ultrafiltration was in the range of 8.9–19.3 mg/L. The efficiency of removal of PAHs from coke wastewater in the process of ultrafiltration equalled 66.6%. Taking into account the initial filtration, the total degree of removal of PAHs reached 85% [30].

Chlorination treatment: Oxidation using chlorine can achieve PAH removal rates similar to those achieved with UV/$H_2O_2$. The indicative removal of benz(a)pyrene, a polyaromatic hydrocarbon (PAH), using chlorination is given, in the AGWR—Augmentation of Potable Water Supplies—as >80% [6]. The modelling in the current study used a PAH removal rate using chlorination of 80% [6].

### 2.3.2. Arsenic, Cadmium, and Lead Removal

Removal of these metals is limited to the wetland, RO, and BAF stages of TTs.

Wetlands: A 2012 literature review by Haarstad et al. (2012) [19] shows the occurrence of more than 500 organic and metallic pollutants in wetlands. The removal of heavy metals is typically reported in the order of 30 to 60%, but it can reach 80 to 90%. A removal value of 30% was used in the modelling for constructed wetlands.

RO treatment: The RO removal of Cd from drinking water with Cd concentrations ranging from 0.02 mg/L to 0.54 mg/L with an average concentration of 0.23 mg/L using a Toray SC 3100 Membrane was reported to be 95% to 99%, with an average removal of 99%. Similarly, the use of a Dow 5K membrane achieved an average Cd removal rate of 98% [20]. The modelling used 98% Cd removal with RO [20].

The RO removal of arsenic (As(III)) has been found to be highly variable. This variability has been attributed to membrane type, matrix effects, and test conditions [20]. The average As(III) removal rates over three separate one-week periods for the Dow 5K membrane were found to be 98%, 75%, and 83%. The modelling used 85% As(III) removal with RO [20].

Ozbey-Unal et al. [21] studied MF-RO treatment of industrial wastewater with a Pb concentration of 1.5 mg/L and found removal efficiency that ranged from 89.3 to 100% for the Pb ion. The modelling used 95% Pb removal for the RO stage in TTA and TTB.

BAF treatment: Arsenic removal using BAF is expected to be higher than removal using RO. The removal of As(III) with BAC has been found to be effective. Pokhrel et al. evaluated As(III) removal from groundwater and found 99% removal using BAC [28].

Biologically active filtration was reported to be able to remove up to 86% of Cd(II) from simulated wastewater with a Cd(II) concentration of 0.2 mg/L (Dong 2018). Adsorption experiments also showed that BAC is able to reduce Pb(II) concentrations by 95% for starting concentrations of less than 0.2 mg/L (Dong 2018).

### 2.3.3. BOD and COD Removal

BOD removal and COD removal were modelled, as these are general indicators of the potential for the generation of harmful disinfection by-products (DBPs) in the advanced oxidation and chlorination stages of TTs. Most of this removal takes place in the early TT stages, thus minimising DBP formation later, in advanced oxidation and/or chlorination.

Wetland treatment: The removal of dissolved organics in free-water-surface wetlands is attributed to phytodegradation, phytovolatilization, phytostimulation, phytoextraction, and microbial degradation. Dissolved heavy metal removal is attributed to precipitation, adsorption, and plant uptake. The removal of undissolved pollutants is attributed to

sedimentation. The performance of constructed wetlands depends upon various factors, such as hydraulic and organic loading rates, pH, dissolved oxygen, temperature, plant species, and growth phase [11]. The selection of literature data on BOD, COD, and heavy metal removal using constructed wetlands was based on general conclusions of literature reviews. A recent literature review on the performance of constructed wetlands in tropical and cold climates concluded that low temperature has the most antagonistic effect on the performance of constructed wetlands [21]. Free-water-surface constructed wetlands were estimated to exhibit around 89% BOD removal and 72% COD removal. These COD and BOD removal values were used in the modelling. Considerably higher BOD and COD removal rates were reported for constructed wetlands (98% and 97%, respectively) [43], but these figures were not used in the modelling, as the Ethiopian study area of that research is subject to a very different climate from that of the study area of the current research. The Ethiopian study compared the performance of constructed wetlands and natural wetlands and found that natural wetlands yielded lower BOD and COD removal (92% for both BOD and COD).

RO treatment: The modelling used 99% for COD and BOD removal using RO. RO has been found to be very effective in the removal of COD and BOD. Jadhao et al. investigated the RO and UF treatment of hospital wastewater that contained 200–235 mg/L COD, 95–115 mg/L BOD and found that the percentage removal efficiency rates of COD and BOD were more than 99% with RO [22].

$O_3$/BAF treatment: The removal of COD and BOD using ozone alone at full scale has been found to be highly variable. Martinez et al. used ozone to treat secondary effluent and achieved 8% to 88% removal of COD, with most of the results being between 10% and 20%, and ~0% to 68% removal of BOD, with most of the results indicating ~0% BOD removal [37]. Similarly, in the case of COD, Zanacic et al. found that two different full-scale $O_3$/BAF plants achieved different COD removal rates [38]. One achieved ~50% COD removal, and the other, ~20% removal. Less variable results were achieved by He et al. [26]. The study investigated the combined use of ozone and biological aerated filtration to treat textile effluent containing ≤110 mg/L COD and ≤30 mg/L BOD. A 20–25 mg/L ozone dose with 3.3 hr hydraulic retention time and an air-to-water flow ratio of 6 was found to result in approximately 64% COD removal and 67% BOD removal [26]. The modelling used 64% COD removal and 67% BOD removal for the $O_3$/BAF combination.

UF treatment: Ultrafiltration has also been found to be able to retain large percentages of COD and BOD. The percentages of removal efficiency of COD and BOD from hospital wastewater using a tight UF membrane with 0.01 μm pore size, approximately equating to a 1 kDa molecular-weight cutoff [44], were found to be more than 99% for RO and more than 97% for UF [22]. For stormwater treatment using a membrane with a 50 kDa molecular-weight cutoff, however, average rates of UF removal of COD and BOD of 42% and 66% were observed [39]. Ultrafiltration does not provide a physical barrier for the separation of dissolved salts and heavy metals, as these are smaller than the membrane pores [31]. The modelling in the current study used 97% COD and BOD removal with UF [22], as this was achieved with a UF membrane with a pore size closer to that of the literature study used to estimate PAH removal [30] (see Table 3).

UV and UV/$H_2O_2$ treatment: The modelling in the current study used conservative BOD and COD removal values of 0% with UV alone and UV/$H_2O_2$. The UV and UV/$H_2O_2$ treatment modelling in the current study was limited to the fluence doses adopted of 12 mJ/cm$^2$ for treatment with UV alone and 800 mJ/cm$^2$ for treatment with UV/$H_2O_2$ [12]. BOD and COD removal is expected to be very low at both these fluence levels. Muhammad et al. [24] found approximately 5% BOD and COD reductions from raw textile bath wastewater for UV alone and approximately 5% COD removal and 15% BOD removal with UV/$H_2O_2$ treatment using irradiance of 5 mW/cm$^2$ (254 nm) for 5 min exposure time, equating to the UV fluence of 1500 mJ/cm$^2$. The maximum reported BOD and COD removal rates were approximately 35% BOD and 25% COD for UV alone and approximately 45% BOD and 30% COD for the UV/$H_2O_2$ treatment after 25 min of expo-

sure at 5 mW/cm$^2$, equating to the fluence of 7500 mJ/cm$^2$. Similarly, Rubio-Clemente [25] found little or no TOC removal from natural water using irradiance of 0.46 mW/cm$^2$ for 5 min, equating to 138 mJ/cm$^2$. TOC removal rates of approximately 20% were achieved using 10 mg/L H$_2$O$_2$ and irradiance of 0.46 mW/cm$^2$ for 30 min, equating to approximately 800 mJ/cm$^2$.

Chlorination treatment: The modelling in the current study used conservative chlorination treatment BOD and COD removal values of 30% [32] and 10% [33], respectively. The strong oxidizing ability of chlorine decreases the amount of residual organic substances and thus can decrease BOD in the effluent. The net effect on BOD and COD is dependent on the chlorine dose and contact time. Chlorine doses up to 5 mg/L added to water from treated sewage and allowed to react for 30 min were found to decrease BOD, but when 30 mg/L or more of chlorine was added, the organic matter was progressively decomposed to low molecular weights, causing BOD and COD to double in the effluent [45]. Another study by Ishihara et al. [32] on chlorination of secondary effluent found that higher percentage removal rates were achieved with chlorination in samples with higher starting BOD. When three different samples with starting BOD concentrations of ~30, 21, and 13 mg/L were treated to a residual chlorine concentration of 5 mg/L, they exhibited BOD reductions of 67%, 38%, and 20%, respectively. Ustun et al. treated industrial wastewater with 25 mg/L COD and 1–3 mg/L NaOCl (0.3 to 1.2 mg/L free chlorine) for 30 min, giving rise to 10% removal of COD [33].

### 2.4. Cost Estimates

The relative cost of the TTs for the treatment of stormwater was assessed on the basis of the results of a study by the WaterReuse Research Foundation on the cost of the treatment of wastewater using RO-based and O$_3$/BAF-based treatment trains similar to those proposed in the current study [46,47]. The study identified the RO-based approach, which is extensively used in California and internationally, as the "gold standard" for potable reuse and compared its cost to that of the O$_3$/BAF approach used in the eastern United States. The costed RO-based TT scenario consisted of MF, RO, and UV/H$_2$O$_2$. The costed O$_3$/BAF scenario consisted of coagulation, O$_3$, biologically active carbon (BAC), granular activated carbon (GAC), and UV disinfection. Neither scenario featured chlorination, and both had product water going to a raw water reservoir to be later treated to potable standards by a water treatment plant. The study used triple-bottom-line accounting considering direct financial costs, and upstream and downstream environmental and social factors: greenhouse gases and air emissions from electricity and chemical use, and transport and disposal of waste. The cost-per-treatment plant capacity data from the study were here used to estimate the capital and operating costs of a plant to treat the stormwater generated in the study area of the west of Melbourne.

## 3. Results and Discussion

### 3.1. Microbial Removal

The expected total bacterial, viral, and protozoan pathogen reductions for the TTs based on the individual default log credit LRVs for each treatment stage are shown in Table 4.

The data in Table 4 indicate that TTA is the only train that can meet the two most conservative LRTs. The other trains fall short with regards to *Cryptosporidium* removal (TTB, TTC, and TTD) and with regards to virus removal (TTB). TTB, TTC, and TTD are expected to yield unsafe drinking water for more conservative (1:10 wastewater dilution) simulated wastewater (*Cryptosporidium* LRT: 8.5).

It can be argued that the AGWR's LRTs for wastewater and Schoen et al.'s [10] 1:10-diluted simulated stormwater LRTs are too conservative. If Schoen et al.'s [10] less conservative LRTs (1:1000 wastewater dilution) were adopted, only TTD would be expected to yield unsafe drinking water. Again, the treated water would not be expected to provide protection against *Cryptosporidium* infection for this simulated stormwater.

Table 4. Estimated treatment train pathogen log reductions.

| | Bacteria (*Campylobacter*) | Viruses (Adenovirus) | *Cryptosporidium* |
|---|---|---|---|
| TTA | 14.2 | 12 | 10 |
| TTB | 14.2 | 9 | 8 |
| TTC | 15.2 | 14.5 | 8 |
| TTD | 15.2 | 11.5 | 6 |
| Default LRTs for wastewater [1] | 8.1 | 9.5 | 8.0 |
| Conservative literature LRTs for stormwater [2] | 8.2 | 8.9 | 8.5 |
| Less conservative literature LRTs for stormwater [3] | 6.2 | 6.9 | 6.5 |

Note: [1] Australian Guidelines for Water Recycling [5], conservative LRTs. [2] Schoen et al., 2017 [10], conservative LRTs: for stormwater with 1:10 dilution of raw municipal wastewater. [3]. Schoen et al., 2017 [10], less conservative LRTs: for stormwater with 1:1000 dilution of raw municipal wastewater.

It is noteworthy that the *Cryptosporidium* density in raw wastewater was estimated to be between 0.3 and 50,000 oocysts/L (Soller 2017), so the 1:1000 simulated stormwater adopted by Schoen et al. [10] could have as many as 50 oocysts/L. The AGWR provide a collation of Australian stormwater quality data from the literature and derive summary statistics. The derived 95th percentile *Cryptosporidium* density for stormwater is given as 546 oocysts/10L (NRMMC, EPHC, NHMRC 2009). The ADWG do not set a guideline value for *Cryprosporidium* but state that if such a guideline value were established, it would be well below 1 organism per litre [17].

Soller et al. evaluated the microbial risk associated with the DPR of wastewater using TTs and LRVs similar to those used in the current study [12]. The potable reuse TTs used by Sollet et al. [12] are consistent with the recommendations of a US expert panel (National Water Research Institute expert panel) [9]. A brief summary of the cumulative annual risk of infection for TTs corresponding to those in the current study is shown in Table 5. The two TTs with an AOP stage, corresponding to TTA and TTC in the current study, were found to deliver a risk of infection considerably lower than the maximum acceptable risk of 1 in 10,000 ($1 \times 10^{-4}$). The two TTs with UV alone, corresponding to TTB and TTD in the current study, delivered a risk of infection close to the maximum acceptable risk. By far the largest contributor to the cumulative annual risk was found to be the risk of infection by *Cryptosporidium*. The risk of infection from the 1:1000-diluted simulated stormwater of Schoen et al. [10] would be expected to be 1000 times lower than the wastewater risk estimated by Soller et al. [12] for wastewater, indicating that all the TTs, including TTB and TTD, are expected to deliver drinking water with very low risk of infection for this simulated stormwater. The risk of infection from the 1:10-diluted simulated stormwater of Schoen et al. [10] is, however, expected to be lower than but close to the unacceptable risk of infection of $1 \times 10^{-4}$ for TTB and TTD ($5 \times 10^{-5}$ and $4 \times 10^{-6}$, respectively).

Table 5. Cumulative annual risk of infection, data from Soller et al. [12].

| Soller et al. [12] TT | Corresponding Current Study TT | Approximate Cumulative Annual Risk of Infection |
|---|---|---|
| TT1a: MF-RO-AOP-Cl$_2$ in storage | TTA | $2.5 \times 10^{-9}$ |
| TT1b: MF-RO-UV-Cl$_2$ in storage | TTB | $5.0 \times 10^{-4}$ |
| TT3a: O$_3$-BAF-UF-AOP-Cl$_2$ in storage | TTC | $1.2 \times 10^{-8}$ |
| TT3b: O$_3$-BAF-UF-UV–Cl$_2$ in storage | TTD | $3.9 \times 10^{-5}$ |

It is noteworthy that Soller et al.'s [12] study used RO LRVs from a review of the literature rather than default credit values. Literature LRVs between 2.7 and 6.5 were reported by Soller et al. [12] for *Cryptosporidium*. Large *Cryptosporidium* oocysts (~10 μm) should easily be retained by the RO membrane filter, so an LRV of 6 is expected for RO. The low LRVs for RO in the literature indicate integrity loss in some of the literature

studies. This highlights the need for critical control point monitoring and validation of TT performance before water is deemed fit for consumption. The results also highlight that TTs that do not rely on an RO stage (e.g., TTC and TTD) can yield safer product water if they also include advanced oxidation (TTC) after the filtration stage.

### 3.2. Chemical Removal

A major challenge in the selection of an appropriate removal percentage to be used in the modelling for each TT stage was the lack of literature studies on the treatment of stormwater. The lack of literature data on stormwater treatment may be attributed to the already mentioned economic and regulatory challenges that have impeded the consideration of stormwater treatment for potable use. Also, for commonly accepted uses of harvested stormwater, the levels of most contaminants in stormwater are very low, thereby negating the need for treatment.

Due to the lack of data on stormwater treatment, the current modelling relied on literature data on treatment of other polluted water types, such municipal wastewater. This could potentially lead to inaccurate estimates of the likely removal efficiency due to concentration and matrix effects. These effects are difficult to estimate for stormwater due to the high potential variability in the chemical composition of stormwater and the effect of the preceding process stages in the TT. The findings of the current study, therefore, need to be considered broadly indicative of the TT performance and chemical safety of treated stormwater. Rather, these findings are used to highlight the broad strengths and weaknesses of the different process trains. The true safety of the use of treated stormwater can only be accurately assessed with a thorough historical knowledge of the specific stormwater and with laboratory and pilot trials of the TT stages.

A comparison of literature statistical water quality data [5] and wetland water quality from analyses in the current project (average of five samples over 5 consecutive months between November and April) with the Drinking Water Guideline values [17] is made in Table 6. The literature 95th percentile data values were greater than the ADWG values for PAH, As, Cd, and Pb (see Comparison 1). The mean levels of the analysed chemical parameters in the effluent from the wetlands were below the ADWG values for all the analysed parameters except for turbidity (see Comparison 2). The analysed chemical parameters in the effluent from the wetlands were higher than the literature data values with respect to barium, sodium, chloride, total dissolved solids, and pH (see Comparison 3).

The modelling of the removal of a large number of chemicals of concern that are likely to be present in stormwater is beyond the scope of the current work. The modelling was limited to the 95th percentile statistical summary chemical water quality data [5] that were close to or exceeded the ADWG values: polyaromatic hydrocarbons (PAHs), As, Cd, and Pb. Biological oxygen demand (BOD) and chemical oxygen demand (COD) are not featured in the ADWG, and their removal rates were modelled as general indicators of water quality. Details of the literature studies and removal efficiency used to estimate TT process performance with regards to PAH, As, Cd, Pb, BOD, and COD removal are given in the Methods section (Table 2 and Section 2.3.2).

The modelled TT product water composition with regards to the chosen key parameters is shown in Table 7. The chemical parameters in treated stormwater close to or greater than the ADWG values are shown in bold font.

**Table 6.** Comparison of literature statistical water quality and wetland water quality from analyses in the current project (average of 5 samples over 5 months) with drinking water.

| Contaminant | Unit | Literature Stormwater Quality [A] | | | | Wetland Analysis Results (Current Project) | | | Comparison 1 | Comparison 2 | Comparison 3 |
| | | Mean | SD | Median | 95th Percentile | Wetlands, In (Mean) | Wetlands, Out (Mean) | Guideline Values [B] | 95th Percentile Literature Data vs. Guidelines | Wetlands (Out) vs. Guidelines | Wetlands (Out) Mean vs. Literature Data Mean |
|---|---|---|---|---|---|---|---|---|---|---|---|
| **Metal** | | | | | | | | | | | |
| Aluminium | mg/L | 1.19 | 0.6 | 1.07 | 2.29 | 0.2380 | 0.318 | N/A | | | Under |
| Arsenic (III) | mg/L | 0.009 | 0.001 | 0.009 | 0.011 | 0.0012 | 0.0015 | 0.01 | **Over** | Under | Under |
| Barium | mg/L | 0.028 | 0.005 | 0.028 | 0.038 | 0.0576 | 0.0364 | 2 | Under | Under | **Over** |
| Cadmium | mg/L | 0.0198 | 0.0242 | 0.0127 | 0.0606 | <0.0002 | <0.0002 | 0.002 | **Over** | Under | Under |
| Chromium | mg/L | 0.009 | 0.001 | 0.008 | 0.017 | <0.001 | 0.0015 | 0.05 | Under | Under | Under |
| Copper | mg/L | 0.055 | 0.047 | 0.041 | 0.141 | 0.0020 | 0.001 | 2 | Under | Under | Under |
| Iron | mg/L | 2.842 | 1.246 | 2.674 | 5.1 | 1.2680 | 0.87 | N/A | - | - | Under |
| Lead | mg/L | 0.073 | 0.048 | 0.063 | 0.162 | <0.001 | <0.001 | 0.01 | **Over** | Under | Under |
| Manganese | mg/L | 0.111 | 0.046 | 0.103 | 0.197 | 0.1780 | 0.0284 | 0.5 | Under | Under | Under |
| Mercury | mg/L | 0.000218 | 0.000105 | 0.000201 | 0.000411 | <0.0001 | <0.0001 | 0.001 | Under | Under | Under |
| Nickel | mg/L | 0.009 | 0.004 | 0.009 | 0.017 | 0.003 | 0.003 | 0.02 | Under | Under | Under |
| Zinc | mg/L | 0.293 | 0.153 | 0.272 | 0.57 | 0.0072 | 0.0102 | N/A | | | Under |
| **Nutrient** | | | | | | | | | | | |
| Total nitrogen | mg/L | 3.09 | 2.33 | 2.51 | 7.46 | 1.3 | 0.54 | N/A | | | Under |
| Total dissolved nitrogen | mg/L | 3.28 | 2.61 | 2.59 | 8.22 | 0.52 | 0.27 | N/A | | | Under |
| Total phosphorus | mg/L | 0.48 | 0.413 | 0.367 | 1.261 | 0.16 | 0.06 | ID | | | Under |
| Ammonia | mg/L | 1.135 | 1.187 | N/A | 3.281 | 0.200 | 0.400 | N/A | | | Under |
| **Organic** | | | | | | | | | | | |
| PAHs (benzo(a)pyrene) | µg/L | 0.262 | 0.306 | 0.168 | 0.811 | <1 | <1 | 0.01 | **Over** | [a] | [a] |

**Table 6.** *Cont.*

| Contaminant | Unit | Literature Stormwater Quality [A] | | | | Wetland Analysis Results (Current Project) | | | Comparison 1 | Comparison 2 | Comparison 3 |
|---|---|---|---|---|---|---|---|---|---|---|---|
| | | Mean | SD | Median | 95th Percentile | Wetlands, In (Mean) | Wetlands, Out (Mean) | Guideline Values [B] | 95th Percentile Literature Data vs. Guidelines | Wetlands (Out) vs. Guidelines | Wetlands (Out) Mean vs. Literature Data Mean |
| **Physicochemical** | | | | | | | | | | | |
| Chloride | mg/L | 11.4 | 1.05 | 11.35 | 13.2 | 37.4 | 14.4 | 250 [(b)] | Under | Under | **Over** |
| Sodium | mg/L | 10.63 | 2.82 | 10.31 | 15.72 | 39.4 | 23.6 | 180 [(b)] | Under | Under | **Over** |
| Total dissolved solids | mg/L | 139.6 | 17.3 | 138.54 | 169.6 | 214 | 126.2 | 600 [(b)] | Under | Under | **Over** |
| Total organic carbon | mg/L | 16.9 | 3.33 | 16.6 | 22.8 | 6.4 | 4.26 | ID | | | Under |
| BOD | mg/L | 54.28 | 45.58 | 42.53 | 140.77 | 4.6 | <2 | ID | | | Under |
| COD | mg/L | 57.67 | 17.22 | 55.75 | 88.72 | 33.3 | 28.5 | ID | | | Under |
| Turbidity | NTU | 50.93 | 40.46 | 40.74 | 127.79 | 7.2 | 5.84 | 5 [(b)] | **Over** | **Over** | Under |
| pH | - | 6.35 | 0.54 | 6.33 | 7.27 | 7.7 | 7.5 | 6.5–8.5 | | | **Over** |

Note: [A] Summary statistics for water quality analysis of stormwater samples from Australian urban catchments (NRMMC, EPHC, NHMRC 2009 [5]). [B] Australian Drinking Water Guidelines (NHMRC, NRMMC 2011 [17]). [(a)] ADWG and literature data mean values lower than limit of detection. [(b)] Aesthetic guideline value

**Table 7.** Modelled treatment train product water composition.

| | PAHs (Benzo(a)pyrene) µg/L | As mg/L | Cd mg/L | Pb mg/L | BOD mg/L | COD mg/L |
|---|---|---|---|---|---|---|
| | **Parameter** | | | | | |
| Feed | 0.811 | 0.011 | 0.0606 | 0.162 | 140.7 | 88.7 |
| TTA | 0.00010 | 0.0011 | 0.00085 | 0.0057 | 0.11 | 0.22 |
| TTB | **0.010** | 0.001155 | 0.00085 | 0.0057 | 0.11 | 0.22 |
| TTC | $5.1 \times 10^{-6}$ | 0.00008 | **0.0059** | 0.0057 | 0.11 | 0.24 |
| TTD | 0.00051 | 0.00008 | **0.0059** | 0.0057 | 0.11 | 0.24 |
| ADWG value | **0.01** | **0.01** | **0.002** | **0.01** | | |

### 3.2.1. PAH Removal

Polyaromatic hydrocarbons are commonly produced through human activities such as the combustion of wood and fossil fuels. They are non-polar, have low solubility in water, and are very persistent in the environment due their high stability. They have toxic, carcinogenic, and mutagenic properties and are thus listed as priority pollutants by the European Union and US EPA. The ADWG only give guideline values for benzo(a)pyrene, as most toxicological studies have been performed on this PAH, and there are insufficient data available to set values for other PAHs. Benzo(a)pyrene is one of the most potent PAH carcinogens. Dibenz(a,h)anthracene has similar carcinogenic potency, while dibenz(a,h)pyrene, dibenz(a,i)pyrene, and dibenz(a,l)pyrene have been estimated to have ten times the carcinogenic potency of benzo(a)pyrene [11].

The modelling reveals a relative weakness of TTB with regards to PAH removal. The PAH concentration in product water with TTB (0.01 mg/L) is expected to be close to the ADWG value of 0.01. The difference in performance between TTA and TTB with regards to PAH removal is due to the expected poor PAH removal with UV treatment compared with advanced oxidation with $UV/H_2O_2$. The PAH level in product water from TTD, which also only uses UV, is low due to the good performance of the $O_3/BAF$ stage (see Table 2).

### 3.2.2. BOD and COD

The ADWG do not provide BOD and COD guideline values, as these are general indicators of organic content in water. They do, however, provide guideline values for individual disinfection by-products (DBPs), which can be formed from organic constituents during disinfection, and emphasise the need to reduce the level of organics prior to disinfection to minimise the formation of DBPs. The Framework for Direct Potable Reuse [9] and the WHO Potable Reuse Guidance for Producing Safe Drinking Water [13] do not give guidance limits for BOD and COD. The WHO Guidance, however, gives the COD guideline limit of 10 mg/L for a case study of a DPR plant in the city of Windhoek, Namibia.

The modelling results indicate that the four TTs are expected to deliver product water with low and very similar organic contents. The bulk of BOD and COD removal with TTA and TTB is expected to take place in the wetlands and in the RO stage, so by the time water reaches advanced oxidation with $UV/H_2O_2$ (TTA) or UV alone (TTB), the organic content in water is very low. This is important because advanced oxidation does not completely mineralize the organic compounds and a significant concern exists regarding the formation of oxidation by-products. For this reason, advanced oxidation is typically used in the final stages of TTs, where the concentration of organics is at its lowest and the expected by-products produced are at their lowest. In TTC and TTD, where $O_3$ oxidation is used in the early stages, the subsequent BAF and UF stages are expected to remove the bulk of the oxidation by-products and any residual organics. Although the overall BOD and COD concentrations in treated wastewater are expected to be similar for all TTs, the DBPs from the $O_3/BAF$-based TTs (TTC and TTD) are expected to be at a higher level in product water, as the COD and BOD levels entering the final chlorination stage are higher for these TTs.

### 3.2.3. Arsenic, Cadmium, and Lead Removal

The modelling results point to a relative weakness in cadmium removal using the $O_3$/BAF TTs (TTC and TTD). The cadmium level in product water (0.006 mg/L) from the two $O_3$/BAF-based TTs, TTC and TTD, is expected to be greater than the ADWG value (0.002 mg/L). Although these are only indicative TT performance estimates, this result points to the need for a monitoring focus on feedwater Cd levels, particularly if implementation of the $O_3$-BAF TT is being considered. It is noteworthy that the Cd levels were below the detection limit in the limited analyses of stormwater (five samples over 5 consecutive months) entering and leaving the wetlands in the current study (see Table 6). The different performance with regards to Cd removal between the $O_3$/BAF-based TTs and the RO-based TTs is due to the difference in the Cd removal ability of the BAF and the RO stages.

### 3.3. Cost Estimates

A comparison of the cost of an RO-based TT (MF-RO-UV/$H_2O_2$) with ocean brine disposal and that of an ozone–BAF-based TT (coagulation, sedimentation, $O_3$-BAC-GAC-UV) is available from a 2014 WaterReuse Research Foundation Report [46], summarized in a USEPA report [14], and shown in Table 8 for a 27.6 GL/yr capacity.

**Table 8.** Costs of alternative treatment trains for a 27.6 GL/yr (20 mgd) facility, an RO facility with ocean outfall (adapted from EPA, 2017 [14] Potable Reuse Compendium, page 5 of chapter 11 all costs in 2022 AUD).

| Cost/Impact | RO-Based TT | Ozone–BAF-Based TT |
|---|---|---|
| Capital cost (millions) | AUD 216 | AUD 164 |
| Annual O&M cost (millions) | AUD 11 | AUD 8 |
| Annual environmental costs (millions) [1] | AUD 3 | AUD 1 |
| Total TBL NPV (millions) | AUD 482 | AUD 312 |
| Cost of water (including environmental costs, AUD/$m^3$) | AUD 0.87 | AUD 0.56 |
| Power consumption (MWh/yr) | 16,000 | 4400 |
| Chemical consumption (dry tons/yr) | 1860 | 1770 |
| $CO_2$ emissions (tons/yr) | 13,400 | 2900 |
| Other air emissions (tons/yr) [2] | 30 | 11 |

Note: [1] Costs associated with meeting environment protection laws, e.g., regarding GHG, noise emissions, and waste disposal. [2] $SO_2$ and $NO_x$.

The relative capital and operational (O&M) costs for different plant capacities are shown in Figure 2.

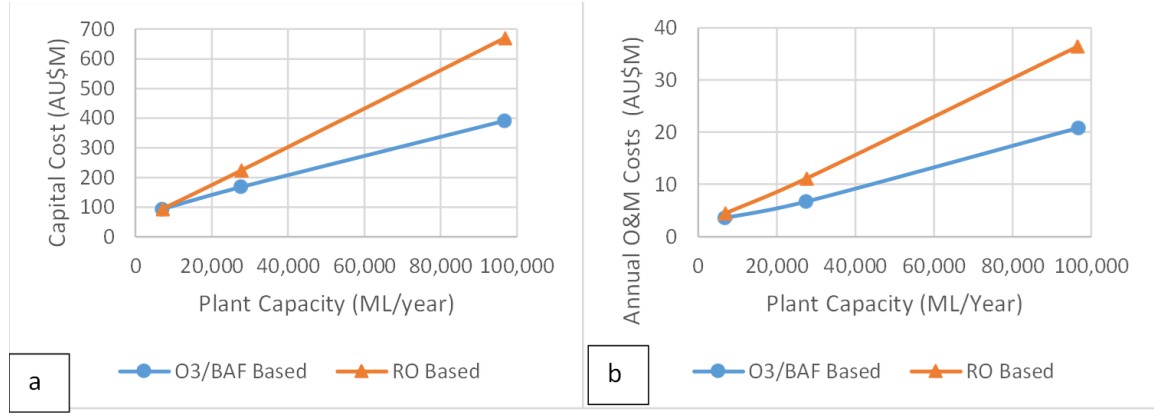

**Figure 2.** Comparison of capital costs (**a**) and O&M costs (**b**) of GAC- and RO-based TTs at different plant capacities; 2022 AUD (adapted from [46]).

Although these TTs have some differences compared with the TTs in the current study, the costings can be used to give a rough indication of the relative costs and environmental considerations associated with the TTs in the current study. Neither literature RO-based nor $O_3$/BAF TTs have the constructed wetlands and the engineered storage buffer that are present in the current study's modelled TTs (see Figure 1). Also, neither scenario featured chlorination, and both had product water going to a raw water reservoir to be later treated to potable standards by a water treatment plant. Additionally, the $O_3$/BAF-based TT in this literature costing does not have a UF stage and has UV alone rather than UV/$H_2O_2$ (see Figure 1). The capital costs of UF, UV, and UV/$H_2O_2$ and the O&M costs of the UF stage stated in the literature costings are shown in Table 9.

**Table 9.** Capital costs (2022 AUD M) of the UF, UV, and UV/$H_2O_2$ TT stages and O&M costs of UF stage from the literature [46]. All figures adjusted for inflation from 2014 to 2022 (29%) with a 2022 USD-to-AUD currency conversion factor of 1.443.

| Cost Type | TT Stage | Plant Capacity (ML/yr) | | |
|---|---|---|---|---|
| | | 6908 | 27,630 | 96,707 |
| Capital | UF | 11.3 | 26.4 | 114.1 |
| | UV | 1.5 | 1.6 | 4.5 |
| | UV/$H_2O_2$ | 5.2 | 18.0 | 43.3 |
| O&M | UF | 2.5 | 5.2 | 18.9 |

These capital costs were used to estimate the indicative capital costs of TTB, TTC, and TTD without wetlands and chlorination. The likely capital cost of the RO-based TT (MF-RO-UV/$H_2O_2$) from the literature costing is the expected cost of TTA without wetlands and chlorination (MF-RO-UV/$H_2O_2$). The likely capital cost of TTD without wetland and chlorination stages ($O_3$/BAF-UF-UV) was estimated by adding the costs of the UF stage to the literature cost of the $O_3$-based TT ($O_3$-BAC-GAC-UV). Similarly, an estimate of the likely capital cost of TTC without wetland and chlorination stages ($O_3$/BAF-UF-UV/$H_2O_2$) was made by subtracting the capital cost of UV and adding the cost of UV/$H_2O_2$ to the likely capital cost of TTD without chlorination ($O_3$/BAF-UF-UV). The approximate likely capital cost of TTB without chlorination (MF-RO-UV) was estimated by subtracting the capital cost of UV/$H_2O_2$ and adding the capital cost of UV to the literature cost of the RO-based TT (MF-RO-UV/$H_2O_2$). The results of these calculations are shown in Figure 3.

Despite these differences, it can be seen that the TTs are estimated to have approximately equal capital costs (AUD 100 M capital cost, 5 M AUD/annum operating and maintenance costs) at low plant capacity (~7000 ML/yr), but these costs are considerably higher for the RO-based TT at higher plant capacities (see Figure 2). The RO-based TT was also estimated to generate considerably more $CO_2$ emissions due to the energy intensity of the RO process (see Table 8).

The available data in the literature costings do not provide O&M costs associated with the individual UV and UV/$H_2O_2$ stages but do provide the O&M costs for UF (see Table 9). It can be seen that the addition of the O&M costs of UF to the literature TT ($O_3$-BAC-GAC-UV) O&M figures in Figure 2b to give an approximate O&M cost for TTD ($O_3$/BAF-UF-UV) brings the total to a higher cost than that of the literature RO-based TT. The use of UV/$H_2O_2$ instead of UV (as for TTC) would further increase the cost due to the added cost of $H_2O_2$.

It is estimated that there was 456 GL of runoff from impervious surfaces in the metropolitan area of Melbourne (approximately 10,000 km$^2$) during 2015–2016 [48]. During the same year, Melbourne's total water use was approximately 430 GL [49]. The metropolitan region of Greater Melbourne has five major waterway catchments [1], and if fully harvested and treated, the runoff is expected to come close to meeting Melbourne's current water needs. The runoff generated from urban areas in the Werribee catchment in the west of Melbourne has been estimated to be approximately 90 GL [1]. Assuming that one

treatment plant is used for the entire volume, the costing estimates in Figure 3 indicate that the TT in the current study at this capacity would have a capital cost, not including wetlands and chlorination, between AUD 600 M and AUD 450 M. The annual O&M is expected to be more than AUD 35 M. The cost of the water is expected to be approximately 0.9 AUD/m$^3$.

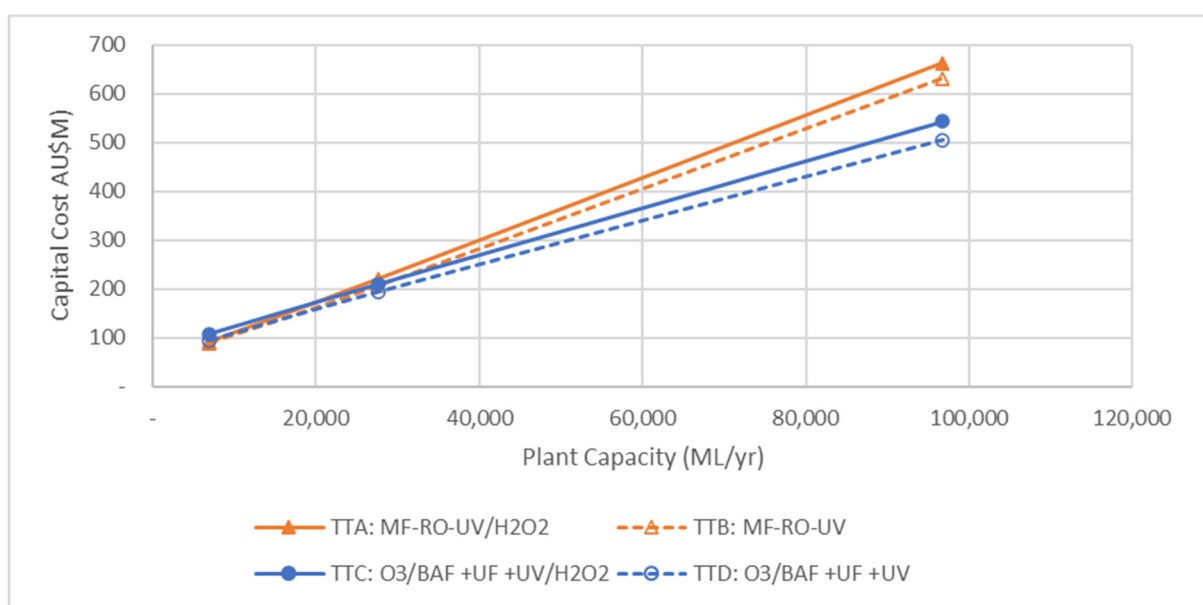

**Figure 3.** Indicative costs of TTs without wetlands and chlorination. All figures adjusted for inflation from 2014 to 2022 (29.1%) with a 2022 USD-to-AUD currency conversion factor of 1.443.

## 4. Conclusions

This study sought to provide information to assist local water authorities in planning for future stormwater management and potable water supply in the high-growth region of the west of Melbourne, Australia, through consultation of the available scientific literature to estimate the likely chemical and microbial characteristics of stormwater, the TT requirements, and the likely costs of treatment to achieve potable standards.

The comparison of the total LRVs for the RO-based TT with the conservative literature LRTs broadly indicated that for these TTs, the inclusion of advanced oxidation (high-dose UV/H$_2$O$_2$) is required for the conservative targets to be met. The conservative LRT for Cryptosporidium removal was not met with low-dose-UV disinfection alone. When compared with conservative literature LRTs, the O$_3$/BAF-based TTs are expected to be deficient with regards to *Cryptosporidium* removal regardless of whether advance oxidation is used. When compared with a less conservative literature LRT, however, the RO-based TTs met or exceeded the targets regardless of whether advanced oxidation was used, and the O$_3$/BAF-based TT without advanced oxidation did not meet or exceed the target.

The modelling of the expected chemical water quality after treatment with these TTs broadly indicated that advanced oxidation is required to ensure PAH removal with the RO-based TTs and that the O$_3$/BAF-based TTs are relatively less effective in cadmium removal, regardless of whether advanced oxidation is used. These results point to the need for a monitoring focus on the feed and product water PAH concentrations if implementation of the RO-based TT without advanced oxidation is considered, and cadmium concentration if the O$_3$/BAF-based TT is being considered.

Based on the available literature, the use of stormwater runoff for potable water supply is likely to be expensive. The treatment of annual stormwater runoff from the Werribee catchment in the west of Melbourne (90 GL) using the RO-based or O$_3$/BAF-based TT in the current study, without wetlands and a final chlorination stage, was estimated to require

a capital cost between AUD 600 M and AUD 450 M, with an annual O&M of more than AUD 35 M.

These findings are broadly indicative, as the specific stormwater quality for the west of Melbourne and TT stage effectiveness in the treatment of stormwater are not currently available. Literature stormwater quality and TT stage effectiveness for wastewater were used in this study. Extensive stormwater quality monitoring for long enough periods to ensure the capture of events that can influence the stormwater composition in the west of Melbourne, such as heavy rain after prolonged dry periods, is required for a more realistic estimate of the effectiveness and safety of the proposed TTs in the generation of potable water from stormwater. The laboratory and pilot testing of TT process stages under the full range of stormwater configurations is also required.

**Author Contributions:** All authors contributed equally to the development of this article. The specific contributions are as follows: conceptualization, P.S., A.K.S., D.N. and S.M.; methodology, P.S., S.M., A.K.S. and D.N.; software and validation, P.S.; investigation, P.S., A.K.S., P.S., D.N. and S.M.; writing—original draft preparation, P.S.; writing—review and editing, A.K.S., P.S., D.N. and S.M.; visualization, P.S., S.M. and A.K.S.; supervision, S.M. and A.K.S.; project administration, S.M., A.K.S. and D.N. All authors have read and agreed to the published version of the manuscript.

**Funding:** We acknowledge the funding received from the Victorian Government of Australia through the Victorian Higher Education State Investment Fund (VHESIF) and Greater Western Water.

**Data Availability Statement:** All the data generated under this study have been included in the publication.

**Acknowledgments:** The authors would like to acknowledge Greater Western Water for their support in selecting the case study site and providing associated data.

**Conflicts of Interest:** The authors declare no conflict of interest.

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
