# Peer review of "Stormwater Treatment Using Natural and Engineered Options in an Urban Growth Area: A Case Study in the West of Melbourne"

_water, doi:10.3390/w15234047_

Round 1
Reviewer 1 Report
Comments and Suggestions for Authors
October 13, 2023
Comments for Stormwater Treatment using Natural and Engineered Options in an Urban Growth Area: A Case Study in the West of Melbourne (2669438)
Journal : Water
Title : Geochemistry indices and biotests as useful tools for the assessment of the degree of sediment contamination by metals
The subject of the study is generally appropriate for Water, according to the journal aim and scope. This topic is interesting and current. However, I have some observations related manuscript and some suggestions to the authors.
Special comments:
· Chapter: Introduction
Line 44-46: The sentence: While the stormwater runoff is expected to be partially mitigated by anticipated drier conditions due to climate change, the water supply challenges are likely to be magnified by drier climatic conditions - citation needed.
Scientific articles in recent years have published data on the increase in rainfall amount, duration, rain extremes occurrences, and a decrease in light rain occurrences in many parts of the world; e.g.:
Wiwoho, B.S., Astuti, I.S., Purwanto, P. et al. Assessing long-term rainfall trends and changes in a tropical watershed Brantas, Indonesia: an approach for quantifying the agreement among satellite-based rainfall data, ground rainfall data, and small-scale farmers questionnaires. Nat Hazards 117, 2835–2862 (2023). https://doi.org/10.1007/s11069-023-05969-0
There are also data on total annual precipitation, which has increased over land areas in the US and around the world:
https://www.epa.gov/climate-indicators/climate-change-indicators-us-and-global-precipitationChapter: 2.5.1. Geochemical indicators
It is therefore unreasonable to expect that climate change can reduce rainwater problems.
· Minor editorial comments:
Line 145, 156 etc. Please, standardise citations throughout the work (Name 2022 or Name et al. 2022).
Line 519: 2.3.2 Please, add "chapter" 2.3.2 in brackets.
Line 592-594 (Table 8) Explanations under the table need to be corrected.
There are also several typos in the manuscript, they have been highlighted in the attached file.

Author Response
Response to Reviewers Comments Manuscript ID: water-2669438 (Revision 1)
Manuscript Title: Stormwater Treatment using Natural and Engineered Options in an Urban Growth Area: A Case Study in the West of Melbourne
Dear Editor
Authors would like to thank reviewers’ for their valuable suggestions/ comments. The manuscript has been revised to incorporate their suggestions and all the edits are in track change in the revised submission.
Responses to all the reviewers’ comments are provided below in this document:
Reviewer 1
R1-Comment 1:
The subject of the study is generally appropriate for Water, according to the journal aim and scope. This topic is interesting and current. However, I have some observations related manuscript and some suggestions to the authors.
Response:
Authors would like to thank reviewer for his/ her positive comment on the submission. We hope this work will provide valuable insights to help urban managers to make better decisions about the management of stormwater resources
Reviewer’s suggestions/ comments have been incorporated in the revised submission as indicated in the responses below to “specific comments” in this document.
Special comments
R1-Comment 2.1:
Chapter: Introduction
Line 44-46: The sentence: While the stormwater runoff is expected to be partially mitigated by anticipated drier conditions due to climate change, the water supply challenges are likely to be magnified by drier climatic conditions - citation needed
Response:
Citation added and additional information about the included citation is provided below:
In accordance with the DELWP (2016) guidelines [2], various climate change impact scenarios for rainfall and PET are described for runoff modelling, as listed below:
- Low-impact climate change: current rainfall and PET factored up by 2.2% and 2.9%,
respectively
- Medium-impact climate change: current rainfall factored down by 2.7% and PET
factored up by 4.7%.
- High-impact climate change: current rainfall factored down by 11.7% and PET factored
up by 5.9%.
(DELWP, 2016) Guidelines [2] for Assessing the Impact of Climate Change on Water Supplies in Victoria Final Report. Available online: https://nla.gov.au/nla.obj-385683871/view (accessed on 25 March 2023).
Following text has been added for further clarification in the manuscript:
It is based on the inference of DELWP (2016) guidelines [2] for the impact on current rainfall and PET changes under low, medium and high climate change scenarios. However, Wiwoho et al. (2023) [3] has highlighted increase in rainfall amount, duration, rain extremes occurrences, and a decrease in light rain occurrences in some parts of the world.
R1-Comment 2.2:
Scientific articles in recent years have published data on the increase in rainfall amount, duration, rain extremes occurrences, and a decrease in light rain occurrences in many parts of the world; e.g.:
Wiwoho, B.S., Astuti, I.S., Purwanto, P. et al. Assessing long-term rainfall trends and changes in a tropical watershed Brantas, Indonesia: an approach for quantifying the agreement among satellite-based rainfall data, ground rainfall data, and small-scale farmers questionnaires. Nat Hazards 117, 2835–2862 (2023). https://doi.org/10.1007/s11069-023-05969-0
There are also data on total annual precipitation, which has increased over land areas in the US and around the world:
https://www.epa.gov/climate-indicators/climate-change-indicators-us-and-global-precipitationChapter: 2.5.1. Geochemical indicators
It is therefore unreasonable to expect that climate change can reduce rainwater problems.
Response:
The above reference (Wiwoho, et al [3]) has been added as reference to incorporate outcome from the above publication as suggested by the reviewer. Please see response for R-1-Comment 2.1.
Minor editorial comments:
R1-Comment 3:
Line 145, 156 etc. Please, standardise citations throughout the work (Name 2022 or Name et al. 2022).
Response:
Authors would like to thank the reviewer for his/ her suggestion. All the references have been updated as per MPDI Water Journal requirements.
R1-Comment 4:
Line 519: 2.3.2 Please, add "chapter" 2.3.2 in brackets.
Response:
Authors will like to thank reviewer for his/her suggestion. “Section” has been added before “2.3.2”.
R1-Comment 5:
Line 592-594 (Table 8) Explanations under the table need to be corrected.
Response:
It has been corrected.
R1-Comment 6:
There are also several typos in the manuscript, they have been highlighted in the attached file
Response:
Authors would like to thank reviewer for his/ her effort in highlighting typos. These have been incorporated.
Reviewer 2 Report
Comments and Suggestions for Authors
In general, the topic and the research work are quite interesting.
There are a few comments for improvement:
The abstract was well-developed.
Since numerous abbreviations are used, I suggest reducing them by removing those which are not replicated many times in the text and also developing a table of abbreviations.
Table 3: What are the numbers in the reference column?
Section 2.4: More elaboration on cost estimate is needed.
Another point is the assumptions and limitations the authors mentioned in their work (e.g. “Due to the lack of data on stormwater treatment, the current modelling relies on literature data on treatment of other polluted waters such municipal wastewater. This can potentially lead to inaccurate estimates of the likely removal efficiency due to concentration and matrix effects.”). Hence, the authors are advised to have an extra section to summarize assumptions and provide limitations of the study for clarifying the future path of the study towards the end of the manuscript.
Author Response
Response to Reviewers Comments Manuscript ID: water-2669438 (Revision 1)
Manuscript Title: Stormwater Treatment using Natural and Engineered Options in an Urban Growth Area: A Case Study in the West of Melbourne
Dear Editor
Authors would like to thank reviewers’ for their valuable suggestions/ comments. The manuscript has been revised to incorporate their suggestions and all the edits are in track change in the revised submission.
Responses to all the reviewers’ comments are provided below in this document:
Reviewer 2
R2-Comment 1:
In general, the topic and the research work are quite interesting.
Response:
Authors would like to thank for his/ her positive comment.
R2-Comment 2:
There are a few comments for improvement:
The abstract was well-developed.
Response:
Authors will like to thank reviewer for his/ her positive comments.
R2-Comment 3:
Since numerous abbreviations are used, I suggest reducing them by removing those which are not replicated many times in the text and also developing a table of abbreviations.
Response:
Authors will like to thank review for his/ her suggestions. Attempt has been made to reduce abbreviations where possible.
R2-Comment 4:
Table 3: What are the numbers in the reference column?
Response:
The numbers have been removed. It was just by a typo.
R2-Comment 5:
Section 2.4: More elaboration on cost estimate is needed.
Response:
Authors would like to thank reviewer for his/ her comment regarding the cost estimation. The Water Reuse Research Foundation (WRRF) study is very detailed and succinct elaboration is difficult. Our costing used cost per plant capacity data from this study as described in the manuscript, and the reader is referred to the specific sections of the WRRF costing for further information. As also highlighted in the document that there is very limited cost data available on the treatment train proposed for the stormwater quality improvement. The information has been collected by contacting some of the equipment manufacturers (info provided in confidence due to business reasons) and some data available for wastewater treatment systems in the literature.
R2-Comment 6:
Another point is the assumptions and limitations the authors mentioned in their work (e.g. “Due to the lack of data on stormwater treatment, the current modelling relies on literature data on treatment of other polluted waters such municipal wastewater. This can potentially lead to inaccurate estimates of the likely removal efficiency due to concentration and matrix effects.”). Hence, the authors are advised to have an extra section to summarize assumptions and provide limitations of the study for clarifying the future path of the study towards the end of the manuscript.
Response:
Authors would like to thank review for his/ her comment on the data on stormwater quality and treatment. As stated, the typical urban runoff (stormwater) quality data has been used in the modelling as the proposed development is currently a greenfield and thus there is no current stormwater quality data available for proposed urban development. Similarly, there is limited examples on stormwater treatment for various end use applications including the efficiency of the treatment processes used. Most of the examples are on wastewater treatment and system efficiencies and thus were considered in modelling the stormwater treatment trains. Authors are of the view that the information included in the document is sufficient in the current form.